# Recruitment of Corticotropin-Releasing Hormone (CRH) Neurons in Categorically Distinct Stress Reactions in the Mouse Brain

**DOI:** 10.3390/ijms241411736

**Published:** 2023-07-21

**Authors:** Krisztina Horváth, Balázs Juhász, Dániel Kuti, Szilamér Ferenczi, Krisztina J. Kovács

**Affiliations:** 1Laboratory of Molecular Neuroendocrinology, Institute of Experimental Medicine Eötvös Loránd Research Network, 1083 Budapest, Hungary; horvath.krisztina@koki.hu (K.H.);; 2János Szentágothai Doctoral School of Neurosciences, Semmelweis University, 1085 Budapest, Hungary

**Keywords:** restraint, predator odor, lipopolysaccharide, osmotic stress, Crh-IRES-Cre, tdTomato, FOS

## Abstract

Corticotropin-releasing hormone (CRH) neurons in the paraventricular hypothalamic nucleus (PVH) are in the position to integrate stress-related information and initiate adaptive neuroendocrine-, autonomic-, metabolic- and behavioral responses. In addition to hypophyseotropic cells, CRH is widely expressed in the CNS, however its involvement in the organization of the stress response is not fully understood. In these experiments, we took advantage of recently available Crh-IRES-Cre;Ai9 mouse line to study the recruitment of hypothalamic and extrahypothalamic CRH neurons in categorically distinct, acute stress reactions. A total of 95 brain regions in the adult male mouse brain have been identified as containing putative CRH neurons with significant expression of tdTomato marker gene. With comparison of CRH mRNA and tdTomato distribution, we found match and mismatch areas. Reporter mice were then exposed to restraint, ether, high salt, lipopolysaccharide and predator odor stress and neuronal activation was revealed by FOS immunocytochemistry. In addition to a core stress system, stressor-specific areas have been revealed to display activity marker FOS. Finally, activation of CRH neurons was detected by colocalization of FOS in tdTomato expressing cells. All stressors resulted in profound activation of CRH neurons in the hypothalamic paraventricular nucleus; however, a differential activation of pattern was observed in CRH neurons in extrahypothalamic regions. This comprehensive description of stress-related CRH neurons in the mouse brain provides a starting point for a systematic functional analysis of the brain stress system and its relation to stress-induced psychopathologies.

## 1. Introduction

Corticotropin-releasing hormone (CRH) is a 41 amino acid neuropeptide, which plays an essential role in the neuroendocrine stress response [1]. Parvocellular neurons in the hypothalamic paraventricular nucleus (PVH) integrate external and internal stress-related information and provide CRH into the hypophyseal portal circulation, triggering ACTH secretion from the pituitary corticotropes, which then stimulate corticosteroid discharge from the adrenal cortex [2,3].

CRH is also involved in mediation of autonomic and behavioral responses to stress. Central (intracerebroventricular) administration of CRH reproduces many of the autonomic- [4], behavioral- [5], neurochemical- [6] and electrophysiological [7] responses seen during stress. 

In addition to PVH, CRH expressing neurons are widely distributed in the CNS, from where they provide ligands for type 1 and type 2 CRH receptors [8] to serve neurotransmitter and neuromodulatory functions [9,10]. While the structure and function of neurosecretory PVH and CRH neurons are well established, much less is known about the stress-related organization of CRH cells outside the hypothalamus.

Visualizing CRH-positive neurons in the brain is rather challenging due to their low basal expression within the cell bodies and the lack of highly sensitive CRH antibodies. To overcome this limitation previous studies have used colchicine treatment to enrich the neuropeptide in the pericaryon via blockade of axonal transport [11]. However, colchicine intoxication questions the physiological relevance of this approach [12].

The introduction of CRH Cre-driver mouse lines [13] in combination with various reporter constructs provides easy tools with which to visualize CRH expressing neurons throughout the CNS [13,14,15]. Previous systematic studies have compared the distribution of natural CRH peptide and of reporter’s expression in three transgenic mouse lines. Among these mouse models, an almost complete overlap between endogenous CRH and the red fluorescence-producing reporter expression was found only in Crh-IRES-Cre;Ai14 tdTomato mice [16].

Cellular immediate-early genes, such as members of the Fos family [17,18], nerve growth factor-induced clone B (*Ngfi-B*) [19] or activity-regulated cytoskeleton-associated protein (*Arc*) [20] are versatile markers of neuronal activation. Their expression is low under resting (no stress) conditions, but they become rapidly and transiently upregulated in response to various external stimuli or intense upstream synaptic activity. Among these activity markers, FOS has been successfully used to reveal hypothalamic and extrahypothalamic circuits involved in mediation of various stress effects [21,22,23,24,25,26,27].

Heterogeneity of CNS circuits activated by different stressors prompted us to investigate whether CRH neurons within and outside of the hypothalamus are also recruited in a challenge-specific manner. Crh-IRES-Cre;Ai9 tdTomato mice were exposed to physical- (ether), physiological- (high salt), psychological- (restraint), immune- (lipopolysaccharide) and innate (predator odor) insults and stress-activated CRH neurons have been mapped by colocalization of FOS and tdTomato.

## 2. Results

To map putative corticotropin-releasing hormone producing neurons throughout the male mouse brain, we used Crh-IRES-Cre;Ai9 reporter mice.

### 2.1. Validation

Because Ai9 mice may express low levels of tdTomato prior to Cre recombination, first, we tested baseline tdTomato signal in Cre-negative controls. Most of the brain areas did not display any red fluorescence; however, within the circumventricular organs, such as the subfornical organ, the median eminence and the area postrema, significant red fluorescence was recognized. These non-neuronal profiles that showed red fluorescence, also displayed green fluorescence and may represent autofluorescence of the tissue. When the signal was checked by anti-tdTomato antibody and DAB chromogen, these areas were negative.

### 2.2. Distribution of tdTomato Positive Neurons in the Mouse Brain

Figure 1A shows the distribution of native tdTomato positive cellular profiles throughout the Crh-IRES-Cre;Ai9 mouse brain. Relatively high density of tdTomato positive profiles was found in the olfactory areas, striatum, pons and medulla. Medium density of TdTomato positive cells was detected in the isocortex, pallidum, thalamus, hypothalamus and midbrain. The hippocampal formation and cortical subplate contained the lowest level of tdTomato expression.

Next, we compared native tdTomato fluorescence observed in Crh-IRES-Cre;Ai9 mice with CRH mRNA distribution published in Allen Brain Atlas (experiment 292, probe RP_Baylor 102704) (Figure 1B). Good correlation between tdTomato density with the reported CRH mRNA raw expression values was seen in isocortex, hypothalamus and medulla. However, a mismatch between tdTomato and CRH mRNA expression was found in the striatum, thalamus, midbrain and pons, where the density of red fluorescent profiles exceeded those of CRH mRNA raw expression. By contrast, in the olfactory areas, hippocampal formation and cortical subplate tdTomato expression was lower than the reported CRH mRNA.

Then, RNAscope in situ hybridization was performed on critical match (PVH) and mismatch (PVTa) areas. This experiment confirmed >80% overlap between tdTomato and CRH expressing profiles in the hypothalamic paraventricular nucleus. By contrast, no CRH mRNA was detected in tdTomato+ cells in the thalamic region (Figure 2).

Finally, we checked whether manipulation of hypothalamic CRH expression affects the appearance of tdTomato in Crh-IRES-Cre;Ai9 mice. Removal of the adrenals results in a significant upregulation of CRH expression in the hypothalamus [28]. Adrenalectomized (ADX) male mice had the same number of tdTomato positive neurons in the PVH than sham-operated controls. The red fluorescence intensity on sections obtained from ADX mice was also comparable to that as seen in sham-operated controls (Appendix A).

Detailed analysis of tdTomato distribution throughout the Crh-IRES-Cre;Ai9 mouse brain is shown on Figure 3. In the hypothalamic paraventricular nucleus, 24.83% of DAPI positive profiles expressed the red fluorescent marker. Similarly, a high ratio (>20%) of tdTomato positive cells was found in the pyriform (PIR) and somatosensory cortices, Barrington’s nucleus (B, the pontine urinary center) and inferior olive (IO). Moderate density (>10% <20%) of tdTomato signal has been detected in the olfactory bulb (MOBopl), in the olfactory tubercle (OT), lateral hypothalamic area (LHA), central amygdala (CEA), nucleus X and lateral reticular nucleus (LRN). Scattered tdTomato labeling with moderate density has been revealed throughout the bed nuclei of stria terminalis and in the fundus of striatum (FS). Within the precomissural nucleus (PRC) a moderate number of tdTomato positive cells were revealed.

### 2.3. Distribution of tdTomato Positive Fibers in the Mouse Brain

One major advantage of Crh-IRES-Cre;Ai9 mice is the visualization of tdTomato positive, presumably CRH fibers and varicosities. Some of these profiles have not been fully resolved by classic immunostaining. In addition to the well-recognized median eminence, we have detected a dense network of td-Tomato-positive varicosities in layer 1 tenia tecta, molecular layers of the piriform and cerebellar cortices as well as close to the substantia nigra, pars compacta (SNc) (Figure 4). According to the Allen Brain Atlas (experiment 297, probe RP_Baylor 253895), most of these areas contain CRH-R1 receptors.

### 2.4. Neuronal Activation

The immediate-early gene, *c-Fos*, as a functional anatomical marker of acute neuronal activation has been used to identify stress-activated neuron population throughout the Crh-IRES-Cre;Ai9 mouse brain.

#### 2.4.1. Controls

There were several brain areas that displayed slight-to-moderate constitutive expression of FOS protein under *no stress* conditions. These areas included: olfactory bulb, prelimbic, infralimbic and piriform cortices, lateral septum, median and medial preoptic nuclei, suprachiasmatic nucleus, paraventricular thalamic nucleus, lateral hypothalamic area, dorsal hippocampus and dentate gyrus, midbrain periaqueductal area, pontine gray, nucleus of the solitary tract and lateral reticular nucleus.

*Intraperitoneal saline injection* (control for ip hypertonic salt group and LPS treatment) resulted in FOS activation (in addition to those seen in non-injected controls) in the cingulate cortex, BNST, posterior PVH, in the thalamic stress-responsive complex (PVT, central medial-, paratenial- and mediolateral thalamic nuclei plus in the medial portion of the reuniens). Structures in which the difference between ip injected and non-exposed groups reached significance were the Mop, PGdm and XII (Appendix A).

Placing animals into a *novel environment* (control for the predator odor exposure) resulted in moderate expression of FOS in all olfactory-related areas, increased FOS intensity in the prelimbic and infralimbic cortices, anterior cingulate area, lateral septum, PVH, ventromedial hypothalamic nucleus, posterior hypothalamic area, thalamic stress-responsive complex, periaqueductal gray and locus coeruleus. Appendix A indicates those areas where the difference is significant.

#### 2.4.2. Stress-Induced Neuronal Activation

Figure 5 shows the proportion of FOS-immunoreactive neurons relative to their respective controls in response to the following stressors: hypertonic salt, LPS, ether exposure, restraint and predator odor.

Twelve tdTomato containing brain areas have been found to be significantly activated by all stressors. These are as follows: the anterior cingulate area (ACA) [29]; claustrum (CLA) [30]; medial amygdalar nucleus (MEA) [31]; bed nucleus of stria terminalis anteroventral division (BSTAv) [32]; geniculate nucleus (G) [33]; anterior hypothalamic nucleus (AHNp) [34]; paraventricular nucleus of the hypothalamus (PVH); lateral hypothalamic area (LHA) [35]; dorsomedial nucleus of the hypothalamus (DMH) [36]; posterior hypothalamic nucleus, ventral part (PHv) [37]; parasubthalamic nucleus (PSTN) [38] and nucleus of the solitary tract (NTS) [21].

Areas that have been exclusively recruited by psychological stressors (restraint and predator odor) include: primary somatosensory area (SSp) [39]; posteromedial thalamic area (PMTH) [40,41]; perifornical nucleus (PeF) [42]; zona incerta (ZI) [43,44,45] and tegmental reticular nucleus (TRN) [46].

The supraoptic nucleus was the only region which has been exclusively activated by systemic (hypertonic salt and LPS), but not by psychogenic stressors.

#### 2.4.3. Stress-Induced Activation of tdTomato Positive Profiles

Next, we analyzed the distribution of stress-induced CRH neurons throughout the Crh-IRES-Cre;Ai9 mouse brain by analyzing colocalization of tdTomato signal (putative CRH) with FOS-immunoreactivity at cellular level. In most of the brain areas, stress-activated neurons were not tdTomato positive. In these areas the rate of colocalization was not significantly different from that seen in respective controls. In general, systemic stressors such as hypertonic salt and LPS injection resulted in fewer double-labeled areas than psychological (restraint, predator odor) stressors, as seen on Figure 6. All five stressors resulted in significant activation of putative CRH neurons in the orbital cortex and anterior cingulate area (ACA). Psychological, but not systemic challenges, activated tdTomato/CRH neurons in dorsal peduncular cortex, in the Barrington’s nucleus and nucleus X. By contrast, putative CRH neurons in the periaqueductal gray were activated by LPS and high salt, not by psychogenic challenges.

Ether exposure, hypertonic salt, LPS and restraint but not predator odor resulted in significant colocalization of tdTomato signal and FOS in the hypothalamic paraventricular nucleus. Detailed analysis of the PVH however, revealed stressor-related heterogeneity in FOS/tdTomato coexpression (Figure 7).

Interestingly, certain tdTomato/CRH-reach brain areas, such as the piriform cortex, central amygdala, BNST and inferior olive display heterogenous, stressor-specific colocalization of tdTomato and FOS markers.

## 3. Discussion

Here we provide a detailed analysis of stress-dependent recruitment of corticotropin-releasing hormone (CRH) neurons in the mouse brain. Putative CRH-producing neurons have been identified using Cre reporter transgenic mice (Crh-IRES-Cre;Ai9) in which a CAG-LoxStopLox-tdTomato construct is integrated into Gt(Rosa)26Sor locus [47]. Because Cre recombinase and CAG promoter are active throughout the whole ontogenesis, it is likely that neurons, which are active during embryonic development, but not in adults, express the marker. Nevertheless, there was a good correlation between tdTomato and CRH mRNA expressions in most of the areas except the thalamus, striatum and midbrain. In these structures, the fluorescent marker expression exceeded that of the mRNA. Chen et al. reported an almost complete overlap of native peptide and the tdTomato reporter in the Crh-IRES-Cre;Ai14 tdTomato mouse, although their analysis was limited to PVH, BNST, amygdala and the hippocampus [16].

Previous studies used Ai14 reporter line [14,16,48], while we found that the Ai9 is more suitable for colocalization studies, due to the moderate expression of the fluorescent marker gene. Importantly, tdTomato expression is independent of strength of CRH expression, because the marker gene is driven by a constitutively active CAG promoter. Indeed, neither adrenalectomy nor acute stress exposure affected tdTomato intensity.

FOS-immunostaining was used as a functional anatomical marker of stress-activated neurons. C-fos is an immediate early gene, which became activated in response to various acute stimuli and peaks at 90–120 min post-challenge [49]. Cell population commonly displaying stress-induced FOS-ir in response to all kinds of stressors is regarded as a core stress-related circuit. Various parts of this core stress circuit have been analyzed in detail in the forebrain and brainstem [25,27,50,51]. Similar commonly activated brain structures have been identified by Pacak and Palkovits [52] using different stressors (immobilization, cold, insulin-induced hypoglycemia, hemorrhage and pain). It should be noted however, that the strength of FOS induction is very heterogeneous even in a commonly activated area of the hypothalamic paraventricular nucleus. Although functional domains of the PVH in mice are not as well distinguishable as in rats, it is likely that the hypophyseotropic neuron population has been ubiquitously activated. Indeed, all stressors used here resulted in significant elevation of plasma CORT (Appendix A).

In addition to the core system, stressor-specific regions have also been revealed. For instance, contribution of olfactory areas was evident in ether and predator odor exposed animals, which were not activated at all in hypertonic salt or LPS-injected animals. By contrast, magnocelluar neurosecretory cells in the supraoptic nucleus displayed significant FOS staining in response to systemic challenges but not to psychological insults. Comparison of FOS patterns in the mouse brain responding to systemic vs. neurogenic stimuli revealed them as highly distinct with more widespread neuronal activation seen after neurogenic stressors. Such differential activation by interleukin-1 (systemic) vs. foot shock (neurogenic) stressors has been described in case of rats [53].

The neurobiological function of CRH goes far beyond the initiation of the neuroendocrine stress cascade. CRH has been highly implicated in mediation of behavioral, emotional and affective aspects of the stress response [54]. For instance, central delivery of CRH results in changes in locomotion, arousal, anxiety, reward, learning and memory [6,55,56]. However, the distribution of extrahypophyseotropic stress-related CRH neurons remains to be precisely located and functionally tested. Colocalization of tdTomato signal with FOS protein was used to identify such CRH neuron population. Among the 95 areas studied, anterior cingulate and orbital/orbitofrontal cortices showed significant colocalization of the two markers in response to all stressors. These cortical areas have been shown to be stress sensitive and implicated in regulation of cognitive functions and affect [57]. In response to psychological stressors CRH neurons in another prefrontal cortical area, the dorsal peduncular cortex, became activated. This cell population has been referred to as master regulator, which connects emotion-related cortical structures with the sympathoexcitatory preganglionic neurons in the dorsomedial hypothalamic nucleus [58]. This circuit has been shown to mediate thermogenic and cardiovascular responses to psychological challenges.

External and internal threats commonly mobilize both the hypophyseotropic and brain CRH systems. CRH containing, median eminence projecting, parvocellular neurons in the PVH initiate the neuroendocrine stress response, while the widespread brain CRH system governs autonomic and behavioral aspects of stress. It should also be noted that these functions are not completely distinct, since glucocorticoid hormones, the end-products of the neuroendocrine stress cascade, also have profound central effects on behavior [59]. Furthermore, a subpopulation of PVH^CRH^ neurons that possess angiotensin I receptor via projections to the lower brainstem orchestrate neuroendocrine and cardiovascular (autonomic) axes [60].

Brain CRH is involved in arousal and affects the sleep–wake cycle. Noradrenergic output from the locus coeruleus (LC) has been implicated as the major regulator of the arousal. LC is innervated by CRH-positive fibers and CRH injections to LC results in sympatomedullary activation. Furthermore, CeA^CRH^ neurons have been revealed as the source of stress-related CRH into LC [61]. The very same LC projecting CRH neurons have also been implicated in stress-induced anxiety, despair and aversion.

More recently, chemogenetic inhibition BNST^CRH^ neurons enhanced fear behavior to predator odor [62], although the target of this CRH-positive neuron population remains to be established. By contrast, systemic IL-1, but not foot shock, resulted in the activation of cells in the same BNST subfield, although the neurochemical nature of these neurons has not been directly addressed [21].

Another interesting finding of this study is the relative weakness of brain CRH recruitment to systemic (LPS and hypertonic salt) challenges compared to stressors with strong psychological components. Indeed, a number of observations suggest the involvement of brain CRH in different psychopathologies. Abnormal levels of CRH were found in the cerebrospinal fluid (CSF) of depressed patients. Melancholic depression was associated with hypersecretion of central CRH and a positive correlation was found between CRH level and the severity of depression. On the other hand, atypical depression is characterized by hypoactive brain CRH activity and suppressed HPA axis. By contrast, PTSD patients have hyperactive central- and blunted neuroendocrine CRH activities [63]. It is very likely that CSF CRH originates from the cortical CRH neurons; however, further functional studies are needed to clarify the recruitment of central CRH neurons in these psychopathological disorders.

In summary, our present comprehensive analysis on the stress-dependent recruitment of brain CRH neurons provides starting points for functional characterization brain stress system and for development of new targets to treat stress-related mental disorders.

## 4. Materials and Methods

### 4.1. Animals

Crh-IRES-Cre (B6(Cg)-Crhtm1(cre)Zjh/J; stock number: 012704) and Ai9 mice (B6;Cg-Gt(ROSA)26Sortm9(CAG-tdTomato)Hze/J; stock number: 007905) were purchased from the Jackson Laboratory and the lines were maintained as homozygotes at the Transgenic Facility of Institute of Experimental Medicine. The Crh-IRES-Cre;Ai9 reporter mice were generated by crossing these homozygote pairs. In F1 heterozygotes Cre-mediated recombination resulted in tdTomato fluorophore expression in CRH neurons. In our experiments adult male mice (10–12 weeks of age) were used. Animals had ad libitum access to standard laboratory chow and water and were kept under standard circumstances temperature: 21 ± 1 °C, humidity: 65%, light: 400 lx with 12L-12D cycle with the light on at 07:00 am hour light-dark phase.

Some Crh-IRES-Cre;Ai9 male mice were deeply anesthetized with ketamine-xylazin (intraperitoneal injection, ketamine, 83 mg/kg; xylazine, 3.3 mg/kg) and adrenalectomized (ADX) from dorsal approach. ADX and sham-operated animals were sacrificed one week after surgery.

All experiments were complied with the ARRIVE guidelines and performed in accordance with the guidelines of European Communities Council Directive (86/609 EEC), EU Directive (2010/63/EU) and the Hungarian Act of Animal Care and Experimentation (1998; XXVIII, Sect. 243/1998). All procedures and experiments were approved by the Animal Care and Use Committee of the Institute of Experimental Medicine, Hungarian Academy of Sciences (permit number: PEI/001/29-4/2013).

### 4.2. Acute Stress

Crh-IRES-Cre;Ai9 mice were caged individually one day prior to stress exposure. Animals were exposed to ether (*n* = 6), hypertonic salt (*n* = 6), lipopolysaccharide (LPS) (*n* = 4), restraint (*n* = 6) or predator odor (*n* = 3) stress. Tests were carried out in a separate experimental room in the early light phase of the day. After stress animals were placed back in their home cage until sacrifice.

#### 4.2.1. Ether Inhalation

For ether stress, mice were placed into a glass chamber saturated with diethyl-ether vapour. As animals became anesthetized (1–1.5 min), they were removed from the chamber and the ether exposure was maintained for total of 5 min using a nose cone with ether-soaked cotton [49].

#### 4.2.2. Hypertonic Salt

Mice were intraperitoneally injected with 1.5 M NaCl solution (0.6 mL/kg body weight) [64].

#### 4.2.3. Restraint

Mice were placed in 50 mL Falcon tubes, in which holes were cut at the end and along the sides to prevent overheating of the animals. Mice were captured in the tubes for 30 min. To achieve a comparable degree of restraint, packing with paper towels at the rear was used. This procedure minimized the space around the animal, prevented them from turning and provided stressful stimulus without being harmful.

#### 4.2.4. Lipopolysaccharide Injection

Lipopolysaccharide (LPS, serotype: 0111:B4, Sigma L4391) was dissolved in sterile pyrogen-free saline and administered intraperitoneally at dose of 1 mg/kg (0.6 mL/kg). The dose was selected to induce substantial systemic inflammation and maximal activation of the hypothalamo–pituitary–adrenocortical axis.

#### 4.2.5. Predator Odor

Mice were exposed to a synthetic analog of fox anogenital product 2-MT [65] (2-methyl-2-thiazoline, CAS 2346-00-1, Santa Cruz Biotechnology, Inc.) in a covered transparent plexiglass arena (40 × 20 × 20 cm) under a fume hood. One day prior to the experiment animals were habituated for 20 min to the test box in which an empty Eppendorf tube lid was placed into the corner. On the day of stress, animals were put into the testing box, where they could freely explore the environment for 10 min, then 2 µL of concentrated 2-MT was pipetted onto a filter paper and placed onto the lid inside the cage. The odor exposure was maintained for 10 min.

#### 4.2.6. Controls

For the restraint and ether stressed animals, undisturbed mice were considered as control group (absolute control) (*n* = 10). In case of lipopolysaccharide and hypertonic salt injected animals, controls were injected intraperitoneally with saline solution (0.6 mL/kg) (ip saline control *n* = 5). Control animals for the predator odor group went through exactly the same procedure as the odor exposed ones, except that saline was pipetted onto the filter paper instead of 2-MT solution (Novel Environment group, NE *n* = 4).

### 4.3. Adrenalectomy

Crh-IRES-Cre;Ai9 mice were anesthetized with a ketamine-xylazine cocktail (16.6 mg/mL ketamine and 0.6 mg/mL xylazine-hydrochloride in 0.9% NaCl, 10 mL/kg body weight intraperitoneally-i.p.). Adrenals were bilaterally removed from dorsal approach. After surgery, mice had a free choice to drink water or saline. Sham-operated animals were used as controls. One week after surgery mice were perfusion fixed.

### 4.4. Perfusion and Tissue Processing

At 90 min after the beginning of stress, at the maximum of FOS protein [49], mice were transcardially perfused with saline followed by 70 mL ice-cold fixative (4% formaldehyde in 0.1 M phosphate buffer, pH 7.2) under terminal anesthesia (Nembutal, Ceva-Phylaxia, Budapest, Hungary). Brains were removed and post fixed in the same fixative supplemented with 10% sucrose for 3 h and cryoprotected overnight in 10% sucrose in potassium phosphate buffered saline, KPBS. Four series of coronal sections (25 μm) were cut on freezing microtome. Sections were stored at −20 °C in antifreeze solution (30% ethylene glycol and 20% glycerol in 0.1 M PBS).

### 4.5. Immunocytochemistry

After 30 min KPBS washing, free-floating brain sections were incubated in 2% normal donkey serum (017-000-121, Jackson ImmunoResearch Europe Ltd., St. Thomas Place, UK) in PBS/0.3% Triton X100 at room temperature for 1 h then either in rabbit anti-c-Fos IgG (sc-52 Santa Cruz Biotechnology, Santa Cruz, CA, USA, 1:10,000) or in sheep anti-tdTomato (provided by Cs. Fekete, Institute of Experimental Medicine, Hungary, 1:50,000) at 4 °C overnight. The immunoreaction was visualized by donkey anti-rabbit IgG (Thermo Fisher, Waltham, MA, USA, 1:1000)/Alexa fluor 488 conjugate or by Alexa Fluor 488 donkey anti-sheep IgG/Alexa fluor 488 conjugate (Thermo Fisher, 1:1000 dilution), respectively, at room temperature for 2 h in dark. Sections were mounted onto slides and coverslipped with DAPI Fluoromount-G (SB. Cat. No. 0100-20) as a nuclear counterstain.

### 4.6. RNAscope In Situ Hybridization (ISH)

RNAscope assay was performed using RNAscope Multiplex Fluorescent Reagent Kit v. 2 (Advanced Cell Diagnostics, Newark, CA, USA) according to the manufacturer’s protocol with modifications [63]. After one hour of KPBS washing, tissues were treated with 1% H_2_O_2_ in KPBS for 30 min. After another 30 min washing, sections were mounted on Superfrost Ultra Plus (Thermo Fisher Scientific, Waltham, MA, USA), then dried at room temperature. As the final step of pretreatment, slides were incubated at 60 °C for 60 min. After a further 20 min of washing in Milli-Q (MQ) water, slides were submerged in 4 °C PFA fixative for 2 min. Following another 30 min MQ water washing, they were put into 0.01 mg/mL proteinase K solution (PK000011, Geneaid Biotech, New Taipei, Taiwan) in 1 M Tris/HCl, pH = 8 and 0.5 M EDTA, pH = 8 buffer at 37 °C for 5 min. After MQ water rinsing tissues endured another 2 min 4 °C PFA treatment. After another 30 min of MQ water washing, sections were hybridized with 3-plex mouse positive or negative control or mouse specific Crh probe. Signal amplification and channel development were applied sequentially according to the manufacturer’s protocol. Used probes and dilution of flourophores are listed in Table 1. Sections were coverslipped with DAPI Fluoromount-G (SB. Cat. No. 0100-20) as nuclear counterstain. Images were evaluated using Nikon Ni-E C2+ laser-scanning confocal microscope equipped with a 1.4 NA Plan Apo VC DIC 60× Oil objective.

### 4.7. Imaging, Quantification and Data Analysis

Digital images of the brain sections were captured at 20× magnification in 3D HISTECH Pannoramic MIDI II. slide scanner. Regions of interest (ROI) were outlined on the basis of Allen Brain Atlas and analyzed with NIS Elements Imaging Software 5.21.01. After immunochemistry, DAPI-, tdTomato- and FOS-positive cells were captured and total area was outlined at each region of interest. Density of cellular profiles was defined as the number of DAPI positive profiles divided by area of the specific region. Correlation among the cell density and area size of the region of interest was calculated. Normalized CRH neuron ratio was calculated as number of the tdTomato positive cells divided by the total cell number (DAPI positive profiles) in a specific region × 100. The ratio of activated neurons was calculated by dividing the number of FOS-immunoreactive cells with the total cell count of the specific region × 100. The ratio of stress-activated CRH neurons was calculated as number of double labeled (tdTomato + FOS) profiles divided by total cell count at ROI. Stress-induced changes in cases of FOS and FOS + tdTomato double labeling were expressed as difference compared to respective control values.

### 4.8. Corticosterone Measurement

A separate set of Crh-IRES-Cre;Ai9 mice were singly housed overnight and exposed to different stressors (ether, hypertonic salt, LPS, restraint or predator odor) as above and decapitated at the maximum of stress-induced hormone output [49]. Trunk blood was collected, centrifuged and serum samples were stored at −20 °C. Corticosterone was determined by direct radioimmunassay (RIA) as described [66].

### 4.9. Statistical Analysis

All quantitative data are expressed as group mean ± SEM (standard error of the mean) for each treatment group. Data were analyzed using GraphPad Prism software (version 7; San Diego, CA, USA). Correlation analysis (comparison of DAPI positive cell number to the area size), one-way ANOVA with Tukey’s multiple comparisons test (tdTomato neuron distribution; neuronal activation and colocalization) and unpaired *t*-test (neuronal activation and colocalization analysis) were performed. We considered *p*-value < 0.05 as statistically significant.

## Figures and Tables

**Figure 1 ijms-24-11736-f001:**
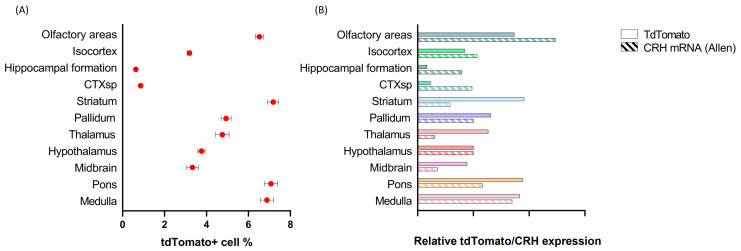
Overview of tdTomato marker expression in the brain of Crh-IRES-Cre;Ai9 mice. (**A**). Density of tdTomato positive cells in major brain areas expressed as number of tdTomato positive cells/number of DAPI positive cells × 100. (**B**). Comparison of tdTomato and CRH mRNA expression (as reported in Allen Brain Atlas). Both markers have been normalized to the hypothalamic values.

**Figure 2 ijms-24-11736-f002:**
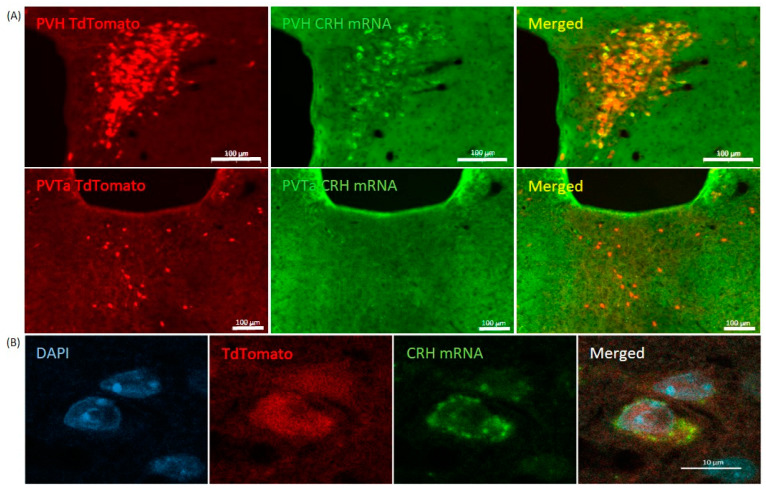
Co-expression of CRH mRNA and tdTomato marker in the hypothalamus and thalamus of Crh-IRES-Cre;Ai9 mice. (**A**). CRH mRNA was visualized by RNAscope and tdTomato was immunostained. The vast majority of tdTomato positive cells display CRH mRNA signal in the PVH. No colocalization of the two markers is seen in the anterior thalamic region. High magnification of a pair of DAPI+, tdTomato+ and CRH mRNA+ cells from the PVH is shown in (**B**).

**Figure 3 ijms-24-11736-f003:**
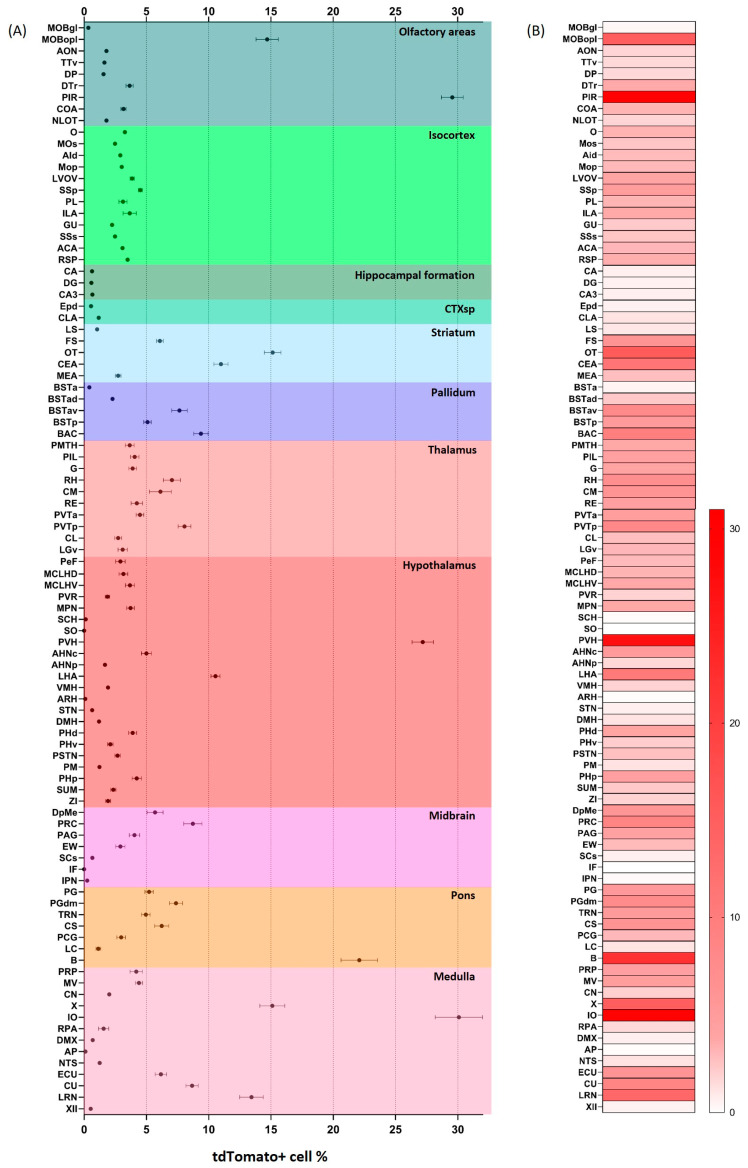
Density of tdTomato expressing neurons throughout the Crh-IRES-Cre;Ai9 mouse. 95 brain areas have been analyzed in detail. (**A**). Mean ± SEM values of tdTomato cell density in the brain of male, non-stressed control Crh-IRES-Cre;Ai9 mice. (**B**). “Heat map” of tdTomato cell density.

**Figure 4 ijms-24-11736-f004:**
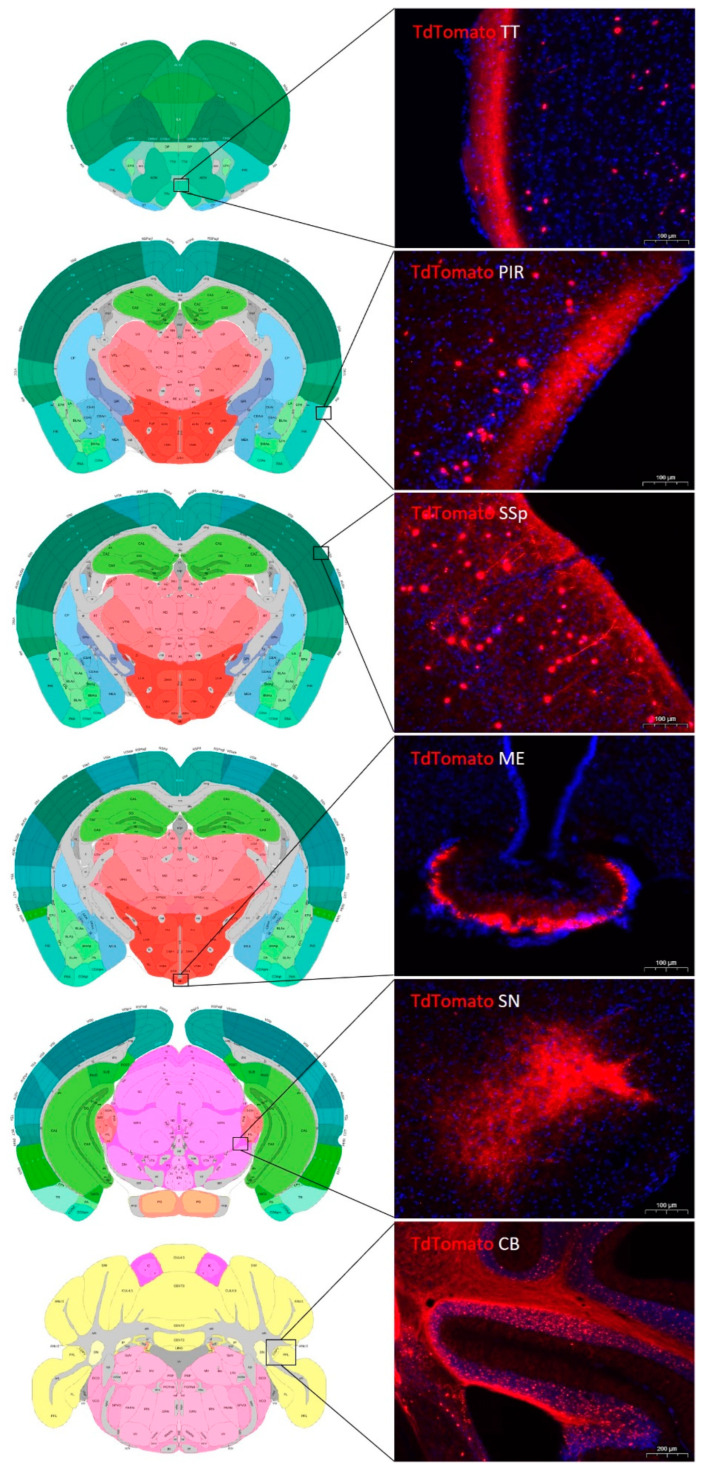
TdTomato immunoreactive fibers/terminals in Crh-IRES-Cre;Ai9 mouse brain. TT-tenia tecta; PIR-Piriform area; SSp-primary somatosensory area; ME-median eminence; SN-substatia nigra; CB-cerebellum. Drawings on the left are from the Allen Brain Atlas.

**Figure 5 ijms-24-11736-f005:**
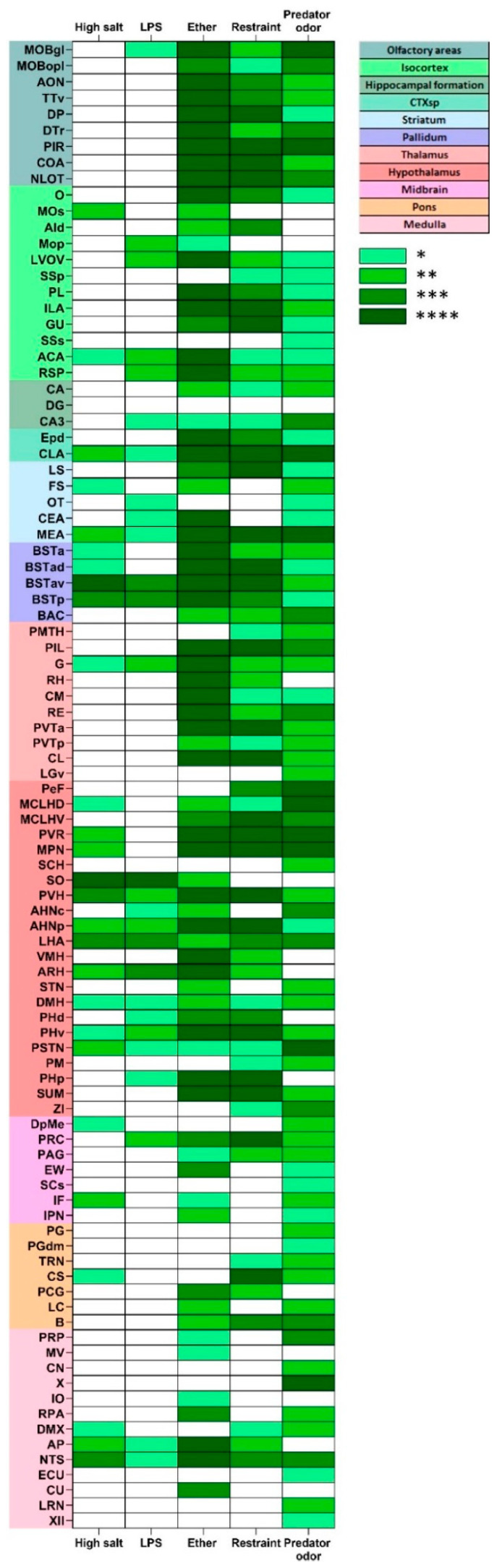
FOS expression in the stressed Crh-IRES-Cre;Ai9 mouse brain. Color-coding refers to the levels of significance between stress-induced FOS density as compared to respective controls. * *p* < 0.05; ** *p* < 0.01; *** *p* < 0.001; **** *p* < 0.0001.

**Figure 6 ijms-24-11736-f006:**
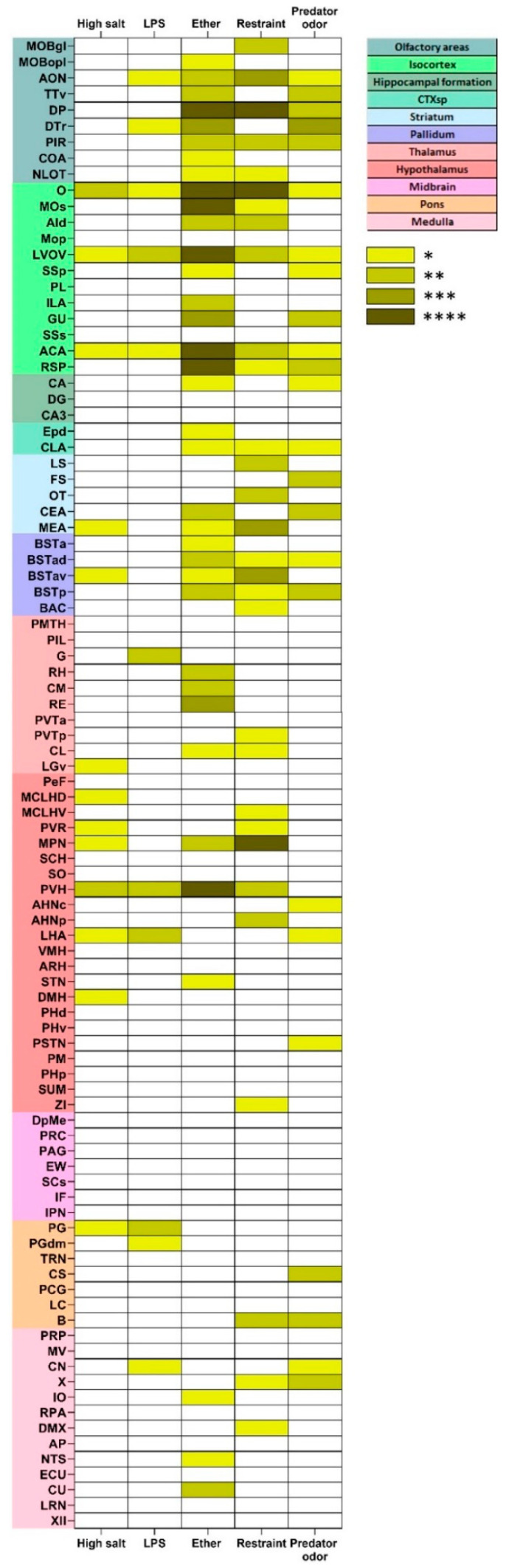
Stress-induced recruitment of putative CRH (tdTomato expressing) neurons in Crh-IRES-Cre;Ai9 mouse brain. Color coding refers to significant differences between cell densities of neurons co-expressing tdTomato and FOS markers as compared to values of respective controls in the same area. * *p* < 0.05; ** *p* < 0.01; *** *p* < 0.001; **** *p* < 0.0001.

**Figure 7 ijms-24-11736-f007:**
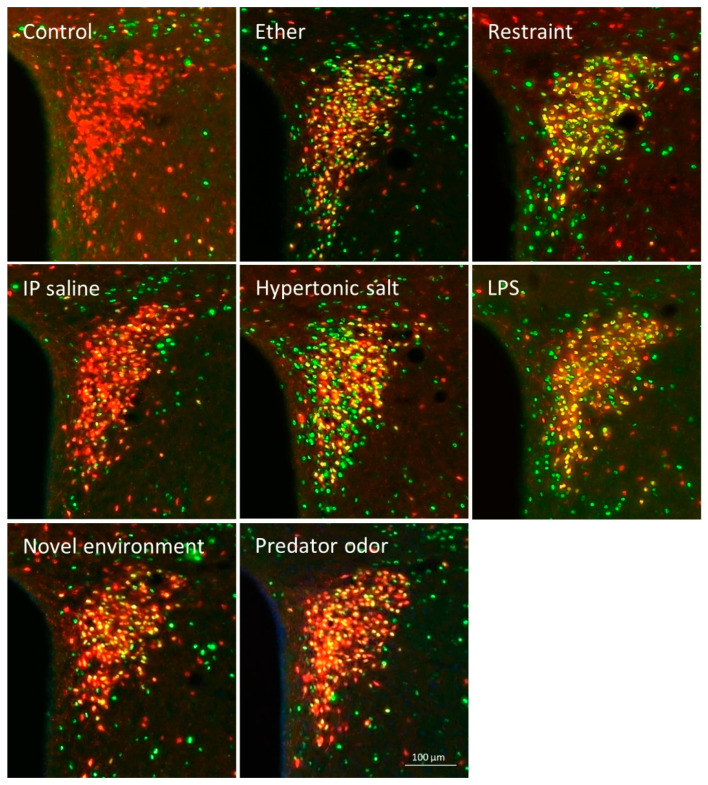
Colocalization of tdTomato (CRH) marker and FOS in the hypothalamic paraventricular nucleus of Crh-IRES-Cre;Ai9 mice in response to different stressors.

**Table 1 ijms-24-11736-t001:** Probes and fluorophore dilutions used in RNAscope measurements.

Target	Catalog Number	Fluorophores	Fluorophore Dilution
Mm *-Crh	316091	Fluorescein Plus TSA	1:3000
3-plex Positive Control Probe-Mm	320881	Fluorescein Plus TSA	1:3000
3-plex Negative Control Probe	320871	Fluorescein Plus TSA	1:3000

* Mus musculus.

## Data Availability

The data presented in this study are available on request from the corresponding author.

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
