# Peer review of "Recruitment of Corticotropin-Releasing Hormone (CRH) Neurons in Categorically Distinct Stress Reactions in the Mouse Brain"

_ijms, 2023, doi:10.3390/ijms241411736_

Round 1
Reviewer 1 Report
Dr. Kovács is a well-known scientist profoundly involved in the circuitries and neurotransmitters/hormones related to the stress response, her carrier was built on that theme. Therefore, the manuscript presented here is well-written, with a very good rationale, methodologically sound, and with very good images. The Discussion chapter is based on very good literature. The results are original and important. Nonetheless, I would point out some issues on nomenclature that I do not agree with, as follows:
1. Please, avoid eponymous such as Ammon's Horn (Hippocampal Formation), or Barrington's nucleus (Pontine Urinary Center). The official Nomenclature Anatomica explicitly recommends avoiding "names".
2. Edinger-Westphal is an exception, just because, even though some authors do refer to the accessory nucleus of the oculomotor nucleus, these names were strongly incorporated in the clinical literature. The actual nomenclature still refers to EW, as EWcp (central projections) and EWpg (EW-pre-ganglionic).
3. I have found two different forms of writing "Locus coeruleus", but, just this one is right. Please, correct the other one "Locus ceruleus".
4. Avoid the acronym "c-Fos", because this form sends us to the very first papers about this proto-oncogene. c-Fos is the cellular form to differentiate from the viral proto-oncogene "v-Fos". c-Fos was primarily described as a gene, therefore, the enterprises think that they are making an antibody against a gene, and this is not true. Please, use the correct nomenclature, such as FOS-immunoreactive (protein - when doing immunohistochemistry), and Fos (in Italics) when referring to the gene.
5. Avoid the use of "paraformaldehyde" when referring to the fixative. Indeed, the fixative is paraformaldehyde in DH2O or some buffer, heated until 60-65 C to dissolve that powder in a liquid, and then we do have the liquid fixative. Therefore, the correct form is: "4% formaldehyde".
Reviewer 2 Report
In this manuscript, Horvath et al. presents a comprehensive analysis of CRH neurons throughout the brain in response to multiple stressors. I find the paper both interesting and well-executed. Although the expression of CRH has been previously reported and reviewed in various papers (for example Deussing and Chen, 2018), the manuscript's valuable contribution lies in elucidating the relative involvement of different brain regions in response to various modalities of stress. This study establishes a solid foundation for future research in the field of stress, paving the way for further advancements in the years to come.
I believe some changes would improve the impact and accessibility of the manuscript. Here are my comments below:
1) The paper offers a comprehensive analysis of CRH neurons in the brain, however some areas, like the substantia innominata, the anterior olfactory nucleus or the olfactory tubercle with reported and reasonably significant CRH expression are not mentioned. Including brain regions that are not in the direct focus of the stress field would be potentially useful for further explorative studies.
2) The authors only show the results of their stress paradigms, however they mention that their controls have increased cFos expression in several regions. Novel context exposure in itself is often considered as a mild stressor and the scruffing that often goes with ip injection could have somewhat similar effects. The authors should show the cFos levels from naïve and control experiments as well.
3) The authors showed a coarse regional correlation between tdTomato expression and CRH mRNA reported in the Allen brain atlas but it is not clear what the graph shows (Fig1b). What does the X axis represent? Why is tdTomato density different from what is shown in Fig1a? Why are there no error bars? Also, the numbers cited from Allen Brain atlas also show discrepancies, for example experiment 292 (RP_Baylor_102704) reports Medulla raw expression value: 2.27; log2: 1.18, but the graph shows different levels (around 1.8-1.9). What is the reason for that?
4) The authors colocalized CRH mRNA and tdTomato in the PVN and in the PVTa. What is the justification or significance of that experiment? Why only those regions were selected for the analysis. What is the colocalization rate in other regions?
5) Similarly, what is the justification of the adrenalectomy experiment? Was it expected that either the number of tdTomato cells or tdTomato intenstity would change? Some explanation would be necessary. Also the statistics are only partially reported here.
6) I believe some structuring of the results in Fig5 and Fig6 would make it much easier to interpret the data. Some additional labeling, similar as shown in Fig3A would help to orientate between the abbreviations.
7) Some parts of the discussion, especially the 5th paragraph (“Neurobiological function of CRH…”) needs more citations.
Author Response
please see the file attached

Reviewer 3 Report
In the present study Horváth et al., first illustrate the activation of the rostral to the caudal extent of the brain following physiological and psychological stressors. Secondly using the Crh-tdTomato mice line illustrated the activation of CRH neurons in the hypothalamic and extrahypothalamic areas. This study advances a basic understanding of the activation of the stress system under various challenges to maintain homeostasis.
There are following concerns remain unanswered:
1) Fig 1B. The authors nicely compared the Crh mRNA with tdTomato cell density and reported that there are discrepancies in co-expression in various brain regions. Therefore, is it rational to use Crh-tdTomato cell line to compare the cFos expression with putative Crh positive neurons? This may either underestimate or overestimate the actual comparison. The authors may want to try Crh RNA scope and cFos co-immunostaining to warrant their findings in the other brain regions where the mismatch is greater.
2) Figures 5 and 6. Although authors reported a low level of cFos expression in PVH following predator order however seems like no co-expression of cFos and Crh-tdTomato positive cells. The authors may want to use another predator order to confirm it.
3) Although the study nicely showed cFos activation in the entire brain following psychological stressors however it remains elusive whether this activation is direct or indirect effects due to a few brain regions activation following psychological stressors.
Round 2
Reviewer 3 Report
All the issues were addressed.